# Effect of Fetal Bovine Serum or Basic Fibroblast Growth Factor on Cell Survival and the Proliferation of Neural Stem Cells: The Influence of Homocysteine Treatment

**DOI:** 10.3390/ijms241814161

**Published:** 2023-09-15

**Authors:** Dražen Juraj Petrović, Denis Jagečić, Jure Krasić, Nino Sinčić, Dinko Mitrečić

**Affiliations:** 1Laboratory for Stem Cells, Croatian Institute for Brain Research, School of Medicine, University of Zagreb, Šalata 3, 10000 Zagreb, Croatia; dpetrovic@genos.hr (D.J.P.); denis.jagecic@mef.hr (D.J.);; 2Department of Histology and Embryology, School of Medicine, University of Zagreb, 10000 Zagreb, Croatia; 3Glycoscience Research Laboratory, Genos Ltd., 10000 Zagreb, Croatia; 4BIMIS—Biomedical Research Center Šalata, School of Medicine, University of Zagreb, 10000 Zagreb, Croatia; 5Laboratory for Neurogenomics and In Situ Hybridization, Croatian Institute for Brain Research, School of Medicine, University of Zagreb, 10000 Zagreb, Croatia; 6Scientific Group for Research on Epigenetic Biomarkers (epiMark), Department of Medical Biology, School of Medicine, University of Zagreb, 10000 Zagreb, Croatia

**Keywords:** cell culture, neural stem cells, homocysteine, β-3-tubulin, DNA methylation, growth factor, epigenetics

## Abstract

In vitro cell culture is a routinely used method which is also applied for in vitro modeling of various neurological diseases. On the other hand, media used for cell culture are often not strictly standardized between laboratories, which hinders the comparison of the obtained results. Here, we compared the effects of homocysteine (Hcy), a molecule involved in neurodegeneration, on immature cells of the nervous system cultivated in basal medium or media supplemented by either fetal bovine serum or basic fibroblast growth factor. The number of cells in basal media supplemented with basic fibroblast growth factor (bFGF) was 2.5 times higher in comparison to the number of cells in basal media supplemented with fetal bovine serum (FBS). We also found that the neuron-specific β-3-tubulin protein expression dose dependently decreased with increasing Hcy exposure. Interestingly, bFGF exerts a protective effect on β-3-tubulin protein expression at a concentration of 1000 µM Hcy compared to FBS-treated neural stem cells on Day 7. Supplementation with bFGF increased SOX2 protein expression two-fold compared to FBS supplementation. GFAP protein expression increased five-fold on Day 3 in FBS-treated neural stem cells, whereas on Day 7, bFGF increased GFAP expression two-fold compared to FBS-treated neural stem cells. Here, we have clearly shown that the selection of culturing media significantly influences various cellular parameters, which, in turn, can lead to different conclusions in experiments based on in vitro models of pathological conditions.

## 1. Introduction

The growth of various types of cells is an obligatory part of the in vitro modeling of any neurological disease. Cell culture techniques frequently incorporate various supplements, as is known, and it has been demonstrated by our group that they can drastically alter cells’ phenotype [1]. Moreover, for some supplements, it has been shown that they can have protective effects. Thus, for example, basic fibroblast growth factor (bFGF) is protective against Ab1–42-induced toxicity [2], while alpha-lipoic acid supplementation was shown to provide protective effects against nanomaterial-induced toxicity [3].

The dominant theory of dementia stresses the importance of age as the only clear significant risk for dementia [4]. However, there is a growing list of evidence that neurodegeneration does not occur only in the aged brain, and it might already occur during fetal development [5,6]. Thus, an attractive hypothesis claims that neurodegeneration, which becomes visible in aged brains, is already seeded during fetal development. Moreover, there is a growing list of evidence linking dementia with some molecules that appear during fetal development, including homocysteine. Homocysteine (Hcy) is a sulfhydryl-containing non-proteinogenic amino acid and a metabolic byproduct produced after transmethylation reactions from methionine (Met) in the cell [7]. Elevated serum Hcy, termed Hyperhomocysteinemia (HHcy), is associated with cardiovascular diseases and Alzheimer’s disease (AD) [8,9], while maternal HHcy has detrimental consequences for the development of the central nervous system [10,11,12]. Interestingly, it was shown that Hcy hinders neural stem cell (NSC) differentiation by inhibiting DNA methylation, which highlights an important link between Hcy and epigenetics [13,14].

With that in mind, here, we investigated a possible link between homocysteine and neurodegenerative changes in immature neuronal cells. We tested an Hcy in vitro model of neurodegeneration by growing cells in three different media: basal medium, basal medium supplemented with bFGF, and basal medium supplemented with fetal bovine serum (FBS). In addition, we tested the expression of MBD1 as a potential link between NSCs’ differentiation and demethylation of CpG islands caused by Hcy.

## 2. Results

### 2.1. Cell Cultures of Precursors of the Nervous System Require Either FBS or bFGF to Prevent Cell Death and Achieve a Satisfactory Level of the Presence of Actin and Nestin Filaments

In order to explore whether different media compositions affect NSCs’ differentiation, we compared the number of cells in the basal medium (BM) (consisting of DMEM/F12 + Glutamax with the addition of 1% Pen/Strep, 1% N2, 2% B27 and 5 mM HEPES) to media enriched with FBS and bFGF (Figure 1A). The results clearly showed that compared to the basal medium, both FBS and bFGF yielded a much more significant survival of the cells, with the bFGF-enriched medium being the most efficient of all of the three tested media (Figure 1B). Moreover, the staining of cells with β-Actin-specific dye highlighted big differences in the morphology of cells in these three conditions, with the basal medium producing the most variable cell appearance.

### 2.2. Medium Enriched by FBS Yields a Significantly Lower Number of Pyknotic Nuclei Compared to bFGF-Enriched Medium and Basal Medium

In order to gain deeper insight into observed differences in cell numbers, we explored nuclear morphology (Figure 2A). After analyzing all possible morphologic variants, we decided to demarcate nuclei into two types: small and more spherous nuclei were recognized as pkynotic nuclei (Figure 2A, white arrows), while all the others were recognized as live nuclei (Figure 2B,C). Detailed image analysis confirmed the difference in sphericity and volume between live and pyknotic nuclei (Figure 2B,C). Comparison of different cell culture conditions revealed that live nuclei in the FBS-enriched group were flatter compared to the BM or bFGF-enriched medium (Figure 2B). In addition, we found that both live nuclei and pyknotic nuclei of the cells grown in FBS-enriched medium had a larger volume compared to the cells grown in other conditions (Figure 2C). We also found that the FBS-enriched medium yields a significantly lower percentage of pyknotic nuclei compared to the BM and bFGF-enriched medium (Figure 2D). At the same time, bFGF-enriched medium had a lower percentage of pyknotic nuclei compared to the cells grown in the BM (Figure 2D). The total nuclei number analysis revealed that bFGF-enriched medium had significantly more nuclei than both BM or FBS-enriched medium (Figure 2E).

### 2.3. Treatment with bFGF Increases the Number of Neuronal Precursors While Marginally Decreasing Cell Proliferation

In order to analyze if cells in the very early and medium stage of maturity are also influenced by bFGF and FBS, we analyzed NSCs on Day 3 and Day 7 of their differentiation (Figure 3A). Comparison of the number of cells between FBS and bFGF revealed that on Day 3, no significant difference was detected (Figure 3B). On the other hand, on Day 7, bFGF-treated NSC cultures had a significantly larger number of the cells (Figure 3B). At the same time, we have confirmed findings from the previous experiment (Figure 2): the bFGF-treated group has a higher percentage of pyknotic nuclei (Figure 3C). To further explore the reason for the high cell number, we co-stained NSCs with the cell proliferation marker EdU. Unexpectedly, this revealed that the cells in FBS-treated group on both days had a strong tendency to divide more than bFGF-treated group; however, statistical significance was not achieved (Figure 3D). In order to analyze whether treatment with FBS or bFGF influences multipotency of our NSCs, we marked them with SOX2. Interestingly, we found no differences for percentage of SOX2 positivity in live nuclei (Figure 3E). In order to better demarcate live nuclei, we decided to add a new criterion based on the presence of the SOX2 nuclear marker in live nuclei. In this independent experiment, we have confirmed findings from our previous experiment (Figure 2): Pyknotic nuclei are indeed significantly more spherical than live nuclei on both days (Figure 3F,G). We confirmed that bFGF induces more spherical nuclei compared to FBS. Interestingly, we found that nuclei sphericity was significantly reduced on Day 7 compared to Day 3 in both FBS- and bFGF-treated groups (Figure 3H). Furthermore, we confirmed that pyknotic nuclei are significantly less voluminous than their live nuclei counterparts (Figure 3I,J). Similarly to the previous experiment (Figure 2), FBS-treated group had significantly more voluminous nuclei, compared to the bFGF-treated group, and the nuclei volume was significantly increased on Day 7 compared to Day 3 for both groups (Figure 3K).

### 2.4. FBS-Treated NSCs Develop into Reactive Astrocytes and Neurons, Whereas bFGF-Treated NSCs Develop into Astrocytes with Quiescent Morphology and Neurons that Increase in Volume over Time

We have observed a significant number of GFAP-positive astrocytes and TUBB3-positive neurons in our cell culture (Figure 4A). Therefore, we decided to explore the volume of each marker normalized to the number of live nuclei. We found that TUBB3 volume was significantly more present in the FBS-treated group compared to the bFGF-treated group on Day 3, whereas TUBB3 volume increased for the bFGF-treated group from Day 3 to Day 7, which could be explained by longer neuron processes (Figure 4C). Considering our culture had a significant number of GFAP-positive astrocytes, we decided to compare how FBS and bFGF influence their morphology (Figure 4A). FBS-treated astrocytes had a significantly greater volume normalized to the number of live cells on both Day 3 and Day 7 when compared to the astrocytes grown with bFGF (Figure 4D). Furthermore, astrocyte volume increased from Day 3 to Day 7 for both FBS- and bFGF-treated astrocytes (Figure 4D). In order to explore whether there is a different amount of MBD1 expression between astrocytes and neurons, we measured the percentage of MBD1-positive live nuclei, and observed that all nuclei were MBD1-positive (Figure 4B).

### 2.5. Homocysteine Decreases Expression of TUBB3 in Both FBS- and bFGF-Treated Immature Neural Cells in a Dose-Dependent Manner without Influence on the Total Number of Cells

Since some data suggested that Hcy might be involved in the pathological changes leading to neurodegeneration, here, we tested how various concentrations of Hcy influence protein expression of TUBB3, which is one of the major structural elements of neuronal projections in immature neural precursors. This influence was tested both in the FBS- and bFGF-treated groups. Day 3 analysis revealed that Hcy decreases the presence of TUBB3 in a dose-dependent manner (Figure 5A), which suggests the potential degeneration of the neuronal structure. Comparison between the FBS- and bFGF-treated groups revealed some differences. While a concentration of 300 µM Hcy already significantly decreased the expression of TUBB3 in the FBS-treated group, this impact was observed only when cells were exposed to a concentration of 1000 µM Hcy in the bFGF-treated group (Figure 5A). Moreover, it was shown that on Day 3, the FBS-treated group expressed much higher levels of TUBB3 than the bFGF-treated group (Figure 5A). Once again, analysis on Day 7 revealed that Hcy reduces the presence of TUBB3 in a dose-dependent manner (Figure 5B). On the other hand, in comparison to Day 3, a significant decrease in TUBB3 expression was observed only when cells were exposed to a concentration of 1000 µM Hcy (Figure 5B). Another distinction was that on Day 7 in the bFGF-treated group, levels of TUBB3 were higher compared to Day 3, when expression was higher in the FBS-treated group (Figure 5B). Interestingly, we found that TUBB3 expression was significantly higher for a concentration of 1000 µM Hcy in the bFGF-treated group compared to the FBS-treated group, which could hint at a protective role of bFGF for TUBB3 expression (Figure 5B).

Since we have detected that Hcy influences the structure of cells by changing TUBB3 expression, we explored whether different Hcy concentrations affect cell death (Figure 6A,E). Interestingly, no significant difference in the percentage of the pyknotic nuclei was found after treatment with Hcy (Figure 6D,H). Furthermore, there was no significant change in the number of live nuclei on Day 3 (Figure 6B). On the other hand, we confirmed that there are significantly more pyknotic nuclei in the bFGF-treated group compared to the FBS-treated group on both days, confirming our previous findings (Figure 6C,G). Unexpectedly, we found a dose-dependent increase in the number of live nuclei on Day 7 for the bFGF-treated group, and a strong tendency toward a dose-dependent increase in the number of pyknotic nuclei, even though significance was lost after correction (Figure 6F,G).

### 2.6. Expression of GFAP Was Much More Strongly Induced by FBS than by bFGF, and It Was Not Influenced by Homocysteine

After we quantified and observed a significant effect of Hcy on neurons (TUBB3-positive cells), we analyzed whether the same effect can be seen in astrocytes. To our surprise, GFAP protein expression exhibited complete resistance to treatment with Hcy (Figure 7A,B). On the other hand, similar to TUBB3 protein expression, on Day 3, significantly higher GFAP protein expression was observed in the FBS-treated group (Figure 7A), while on Day 7, higher GFAP protein expression was observed in the bFGF-treated group (Figure 7B).

### 2.7. Although MBD1 Expression Is Unaffected by Hcy Treatment, Early Neuronal Precursors Treated with FBS Have Higher MBD1 Expression

Since we hypothesized that the effects of Hcy might be linked to expression of epigenetic marker MBD1, we tested how various concentrations of Hcy in various growing conditions would influence it. Although MBD1 protein expression was unaffected by different Hcy concentrations, we found a statistical significance for increased MBD1 protein expression in the FBS-treated group for Hcy control and a concentration of 1000 µM Hcy, in comparison to the bFGF-treated group (Figure 8A). No effect was observed on Day 7 (Figure 8B).

### 2.8. bFGF Significantly Increases SOX2 Expression on Both Day 3 and Day 7, While SOX2 Expression Is Unaffected by Hcy Treatment

Given that FGF activating mutations result in a significant increase in SOX2 protein expression, we decided to analyze SOX2 expression in our model [15]. We have indeed observed a significantly higher protein expression of SOX2 on Day 3 for Hcy control and a concentration of 1000 µM Hcy, with a strong tendency for higher expression in a concentration of 300 µM Hcy in the bFGF-treated group compared to the FBS-treated group (Figure 9A). Furthermore, there was a significant induction of SOX2 expression for bFGF-treated group on Day 7 (Figure 9B). SOX2 expression was completely resistant to Hcy treatment (Figure 9A,B).

### 2.9. Analysis of the DNA Methylation Status of Tubb3 and Sox2 Promoters Revealed the Impact of Treated Media on Epigenetics

Analysis of the DNA methylation of the *Tubb3* promoter on Day 3 revealed that CpG5 was significantly hypomethylated for around 3% in the bFGF-treated group compared to the FBS-treated group, but only for Hcy control (Figure 10B). Furthermore, the average methylation of all six CpG dinucleotides showed a significant hypomethylation in the bFGF-treated group compared to the FBS-treated group on Day 3 (Figure 10C). Unexpectedly, on Day 3, looking at the bFGF-treated group at a concentration of 300 µM Hcy, the average methylation of all six CpG dinucleotides showed significant hypermethylation compared to Hcy control or a concentration of 1000 µM Hcy (Figure 10C). On Day 7, we observed a significant hypomethylation of CpG5 in the bFGF-treated group compared to the FBS-treated group, but only for a concentration of 1000 µM Hcy (Figure 10D). This was further confirmed by the average methylation of all six CpG dinucleotides (Figure 10D). On the other hand, analysis of the DNA methylation of the Sox2 promoter revealed significant changes only on Day 3 of differentiation. There was a significant hypomethylation in CpG2 in the bFGF-treated group compared to the FBS-treated group, but only for the Hcy control (Figure 10G). However, CpG9 was significantly hypermethylated in the bFGF-treated group compared to the FBS-treated group, but only for a concentration of 1000 µM Hcy (Figure 10H).

## 3. Discussion

In vitro models are a very common tool in neuroscience. However, it is known that cell culture protocols are not strictly standardized, which may lead to many still poorly investigated differences. These have been recently reported by our group [1] and others [16].

In this work, we aimed to analyse the influence of two very commonly used supplements, FBS and bFGF, that were added to cell culture in an in vitro model of the neurodegeneration of immature cells in the nervous system, which was prompted by treatment with Hcy.

In order to achieve better cell survival, we found that adding either FBS or bFGF to the basal media increased the cell number and cell survival. Previous studies have confirmed that apoptotic nuclei have a smaller nuclear area [17]. However, it is very hard to differentiate between apoptotic and necrotic pyknosis by analyzing nuclei morphology [18]. Furthermore, toxin-induced pyknotic cell death in cultured cells resulted in significantly reduced nuclei volume [19]. Therefore, we decided to differentiate between pyknotic and live nuclei based on nuclei volume. Interestingly, we found that pyknotic nuclei had a significantly lower nuclei volume and a significantly higher nuclei sphericity compared to live nuclei. Moreover, we found that FBS-treated live nuclei had a significantly bigger nuclei volume and a significantly lower nuclei sphericity compared to bFGF-treated live nuclei. Chen et al. reported that nuclear morphology is regulated by cell shape [20]. Therefore, we assumed that a significantly greater volume of GFAP in FBS-treated NSCs, with a visibly spread out cell morphology (Figure 4A), directly affects nuclear morphology.

Israsena et al. report that FGF2 helps to maintain neural progenitor cells in a proliferative state [21]. On the other hand, NSCs isolated from neonatal rats, passaged as neurospheres and plated as single cells in media composed of 2% N2 and 5% FBS, differentiate equally into GFAP+ astrocytes and TUBB3+ neurons [22].

Fetal bovine serum (FBS) is commonly used in cell biology research because it contains a variety of growth factors, hormones, vitamins, amino acids, fatty acids, and a number of unspecified growth factors necessary for cell survival and proliferation [23]. Our own group sometimes uses FBS when longer survival of cells is needed [24]. On the other hand, it is known that FBS and other animal sera exhibit large batch-to-batch variability, which can have an impact on cellular proliferation, differentiation, and function. Moreover, FBS remains a concern to researchers and clinicians due to unpredictable product variability, clinical risks of adverse reactions due to bovine proteins or disease, and ethical problems linked to its production [25,26].

Basal fibroblast growth factor (bFGF), also known as FGF-2, is a growth factor present in three isoforms (17, 21 and 23 kDa) that induces the proliferation and differentiation of multipotent neural stem cells [27,28,29,30]. Moreover, it has been shown that both LMW and HMW isoforms act protectively against Ab1–42-induced toxicity in cortical astrocytes, whereas only the LMW isoform promotes astrocyte proliferation [2]. Data about possible neuroprotective properties of bFGF in neural stem cells are lacking.

Hcy is reported by Lin et al. to have toxic effect on NSC neurospheres by reducing their diameter and increasing LDH activity in media [14]. On the other hand, Rabaneda et al. reported that even though Hcy did reduce neurosphere size, it did not influence cell viability or cell survival in vitro or in vivo [31]. Wang et al. reported that Hcy induces DNA inter-strand cross-links via oxidative stress, leading to apoptotic cell death [32]. Contrary to that, Maler et al. reported that Hcy concentrations of only above 2 mM induce cell death in in vitro rat astrocytes [33].

We assumed that differences reported in different studies could arise from variations in types of cell cultures, unstandardized media composition, or unrelated methods for analyzing cell death and/or cell survival. Indeed, Lin et al. used NSCs isolated from neonatal rats, which they grew as neurospheres in the growth media (composed of DMEM containing a 2% B27 supplement, 20 ng/mL EGF, 20 ng/mL bFGF, 2 µmol/mL L-glutamine, and 100 U/mL penicillin and streptomycin), and they analyzed neurotoxicity by measuring LDH released into the media [14]. On the other hand, Rabaneda et al. used neural progenitor cells (NPCs) obtained from the subventicular zone (SVZ) of postnatal (P7) mice, which they grew in media composed of DMEM/F-12 (1:1) plus the medium supplement B27, 2 mM glutamine, 2 µg/mL gentamicin, 20 ng/mL EGF, and 10 ng/mL bFGF [31]. However, they measured cell death by counting the pyknotic nuclei of dissociated cells that were plated onto poly-l-ornithine (PLO)-coated eight-well glass slide chambers [31]. Wang et al. used immortalized C17.2 mouse NSCs maintained in Dulbecco’s modified Eagle’s medium (5 mM glucose) supplemented with 10% fetal bovine serum, 5% horse serum, 2 mM glutamine, 100 U/mL penicillin, and 100 mg/mL streptomycin [32]. Interestingly, they observed that Hcy induces NSC apoptosis, as measured using flow cytometry [32]. Besides the fact that these studies vary in their methodology, the effects of Hcy on cell death are even more confounded by the fact that almost every study uses different Hcy concentrations, which have been demonstrated to have an impact on cell survival [33]. Growth media formulations are usually serum-free media used for primary NSC cultures obtained from mice or rats, composed of DMEM containing a 2% B27 supplement, 20 ng/mL EGF, 20 ng/mL bFGF, 2 µmol/mL L-glutamine, and 100 U/mL penicillin and streptomycin with or without 4 mg/L folic acid [14,34]. On the other hand, media used for immortalized cell lines usually contain animal sera, and they are formulated as follows: Dulbecco’s modified Eagle’s medium (5 mM glucose) supplemented with 10% fetal bovine serum, 5% horse serum, 2 mM glutamine, 100 U/mL penicillin, and 100 mg/mL streptomycin; or Minimum Essential Medium (MEM; Gibco, Carlsbad, CA, USA) with 10% fetal bovine serum, 2 mM glutamine, 100 U/mL penicillin, and 100 μg/mL streptomycin, for NE4C cells [32,35].

Since different media compositions affect NSCs differently, we decided to gain deeper insight in Hcy’s effects on immature neural precursors while modifying the basal medium with either FBS or bFGF. To achieve that, we compared the efficiency of the cell culture in terms of cell survival and cell proliferation in the three following conditions: basal medium (BM), BM with added FBS, and BM with added bFGF. We clearly showed that supplementation with bFGF increases the cell number. Additionally, we have found that supplements influence cell morphology. It has been previously reported that astrocytes cultured in serum containing media have higher GFAP levels and reactive morphology [36]. Interestingly, we have observed that just 1% of FBS also leads to the development of reactive astrocyte morphology, compared to bFGF-treated astrocytes, due to a significant increase in GFAP volume (Figure 4A,C).

Our next step was to investigate how Hcy influences immature neuronal precursors. Here, we have found a very clear dose-dependent influence on TUBB3. After analyzing the literature, herein, we propose the possible reasons for the decreased expression of β-3-tubulin following exposure to Hcy.

Hcy leads to disturbed protein degradation rates in cells; more precisely, cell incubation media saturated with Hcy remarkably increased the degradation rates of many protein fractions [37]. One of the possible reasons for this is based on deleterious posttranslational modifications of β-3-tubulin protein, which could in turn lead to its conformational instability and faster protein degradation through the proteasome. This is comparable to the observation that β-3-tubulin mutations lead to its conformational instability and faster degradation by the proteasome [38]. These posttranslational modifications could be a consequence of oxidative stress, S-homocysteinylation or N-homocysteinylation. Hcy in high concentrations acts as an oxidant due to its interaction with the heme proteins of the cell [39,40]. S-homocysteinylation could occur when Hcy directly reacts with β-3-tubulin cysteine residues, mimicking glutathione [41]. Furthermore, Chen et al. have demonstrated that proteins in hela cell lysates undergo unspecific N-homocysteinylation of lysine residues, and that TUBB3 is among them [42].

Another possible reason for the reduction in the protein level expression of β-3-tubulin could be the assumed N-homocysteinylation of β-3-tubulin, which could lead to its subsequent aggregation into amyloid-like protofibrils [43,44]. These aggregates are either cleared by proteasome system or by autophagy, which could lead to cytotoxicity and cell death in the case that the autophagy is unsuccessfully executed [45].

Another mechanism could be due to β-3-tubulin destabilization, resulting from dissociation of N-homocysteinylated tau and MAP1 from microtubules [46]. This process was extensively explored in vitro by Karima et al., who found that N-homocysteinylated human 4R/1N tau caused disassembly of microtubule protofilaments and ribbons [47].

Another important finding was that supplementation with either FBS or bFGF influences how cells react to different Hcy concentrations. On Day 7 of differentiation, there was a Hcy-dose-dependent increase in the live cell number in the bFGF-treated group. Wang et al. reported that NSCs express cystathionine β synthase (CBS), which metabolizes excess cysteine into H_2_S, which induces NSCs’ proliferation [48]. CBS preferentially condensates cysteine with homocysteine, resulting in the production of cystathionine and H_2_S, a process through which toxic Hcy is additionally metabolized [49]. The relevance of CBS in Hcy metabolism is so significant that patients with CBS deficiency exhibit elevated Hcy plasma levels of over 200 μM, compared to 5–15 μM in healthy adults [50]. Singh et al. reported that H_2_S promotes proliferation in mouse NSCs [51]. Therefore, we hypothesize that externally added Hcy reacted with the cysteine already present in DMEM/F12, leading to increased production of H_2_S, which acts to increase NSCs proliferation, explaining the mechanism behind the increased cell number. However, further work is warranted to explore whether this is the exact mechanism. Furthermore, we found a strong tendency toward an increasing number of pyknotic nuclei with increasing Hcy concentrations only for the bFGF-treated group on Day 7 (Figure 6G). However, we found no difference in the percentage of pyknotic nuclei when normalized to all nuclei. We propose that in the immature cells of the nervous system, Hcy acts synergistically with bFGF to further increase cell proliferation, but this process is accompanied by an increase in cell death.

Hcy can exert its toxicity through multiple pathways: epigenetic dysregulation, leading to global hypomethylation; toxic protein modification, mainly via irreversible N-homocysteinylation; direct excitotoxic binding to NMDA receptors; and oxidative stress [52,53,54,55]. Irreversible and toxic N-homocysteinylation of tau and MAP1 leads to dissociation from β-tubulin [46]. Inhibiting DNA methylation hinders NSC differentiation, highlighting the significance of epigenetics in this process [13]. Lin et al. demonstrated that Hcy-induced DNA hypomethylation may be caused by a reduction in DNMT activity, which is regulated by cellular concentrations of S-adenosylmethionine (SAM) and S-adenosylhomocysteine (SAH) [14].

Since we have observed a significant and specific dose-dependent decrease in TUBB3 protein level expression as a consequence of increasing Hcy concentrations, we decided to explore whether this could be due to changes in DNA methylation of the *Tubb3* promoter. Cao et al. reported that PAX3 expression decreases with NSC differentiation, while TUBB3 expression increases; furthermore, they found that PAX3 directly binds to the *Tubb3* promoter [56]. Therefore, we hypothesized that the *Tubb3* upstream promoter could be regulated by DNA methylation, and in order to narrow down our search, we focused on the region containing the MBD1-binding motif TGCGCA [57]. Recently, it has been reported that the *Tubb3* promoter is not regulated by DNA methylation during neural stem cell differentiation; however, this group sequenced a segment of 500 bp upstream of the *Tubb3* transcription start site, and found very low average DNA methylation in this segment [58]. In our study, we focused on a smaller region of 77 bp located about 1000 bp upstream of the *Tubb3* gene transcription start site, where we found an average DNA methylation of six CpG dinucleotides of about 60%. Therefore, we hypothesize that this region could have a regulatory role in *Tubb3* gene transcription, and that CpG5 dinucleotide is affected by different media supplementation. We found that Day 3 bFGF-treated NSCs have lower DNA methylation of CpG5 and average methylation of the region in Hcy control, whereas on Day 7, the same occurs, but only for a concentration of 1000 µM Hcy. We propose that either bFGF or FBS supplementation has greater effect on immature NSCs, whereas Hcy tends to exert its effect on more developed NSCs.

Since we also found that significant changes in SOX2 protein expression levels depend on supplementation with either FBS or bFGF, we decided to examine whether this could be a consequence of different DNA methylation levels in the *Sox2* promoter. Interestingly, we found a very low level of methylation for this promoter, of just 1–3% (Figure 10G,H). Low levels of DNA methylation in the *Sox2* promoter in stem cells are expected; however, small changes in DNA methylation can have large biological effects. We conclude that the detected changes in DNA methylation of *Sox2* promoter have biological effects. Bunina et al. provided evidence that in mice, the *Sox2* promoter 1.5kb upstream of TSS is heavily regulated during mESC differentiation into neurons by DNA methylation, among other epigenetic regulatory mechanisms [59]. Furthermore, Miyagi et al. have confirmed that Sox-2 regulatory region 1 (SRR1) and SRR2 regions are specifically active in multipotent NSCs [60]. Given that the *Sox2*gene is heavily regulated by DNA methylation of its promoters upstream, we observed that this regulatory region extends for an additional 100 bp upstream (Figure 10F).

## 4. Materials and Methods

### 4.1. Isolation and Passaging of Neural Stem Cells

For the purposes of this research, we used neural stem cells (NSCs). NSCs were isolated from telencephalic wall of 14.5-day-old C57/BL6 albino mice embryos. Pregnant C57Bl6 albino mice were sacrificed by cervical dislocation, and 14.5-day-old embryos (E14.5) were removed and dissected as previously reported [61]. Briefly, the telencephalic walls were isolated, mechanically dissociated, and incubated in Accutase (A1110501, Life Technologies Corporation, New York, NY, USA) for 20 min at 37 °C. After a wash, cells were resuspended in proliferation medium consisting of DMEM/F12 + Glutamax (31331-093, Thermo Fisher Scientific, New York, NY, USA) with addition of 1% N2 (17502-048, Thermo Fisher Scientific, Waltham, MA, USA), 1% Pen/Strep (Penicillin/Streptomycin, 5000 U/mL, 15070063, Thermo Fisher Scientific, Waltham, Massachusetts, USA), 2% B27 (17502, Thermo Fisher Scientific, Waltham, MA, USA), 20 ng/mL EGF (Epidermal growth factor, PMG8041, Thermo Fisher Scientific, Waltham, MA, USA), 10 ng/mL bFGF (fibroblast growth factor basic, PMG0035, Thermo Fisher Scientific, Waltham, MA, USA), and 5 mM HEPES (H0887, Sigma-Aldrich, Saint Louis, MO, USA). They were cultivated in suspension, in which they formed structures called neurospheres. Neurospheres were dissociated after 4 days, and on every second day, 20 ng/mL of EGF and 10 ng/mL of bFGF was added to promote neurosphere growth.

### 4.2. Differentiation and Treatment of Neural Stem Cells

To start differentiation, cells were dissociated with Accutase and plated at 1 × 10^5^ cells/mL on poly-D-lysine (PDL, P6407, Sigma-Aldrich, Saint Louis, MO, USA) in a final concentration of 50 μg/mL, and in laminin (L2020, Sigma-Aldrich, Saint Louis, MO, USA) in a final concentration of 10 μg/mL pretreated plates, where complete proliferation medium was replaced with basal medium (BM) consisting of DMEM/F12 + Glutamax (31331-093, Thermo Fisher Scientific, New York, NY, USA) with addition of of 1% Pen/Strep, 1% N2, 2% B27 (17502, Thermo Fisher Scientific, Waltham, MA, USA) and 5 mM HEPES. Basal medium was supplemented with 1% FBS (fetal bovine serum, 10270106, Thermo Fisher Scientific, Waltham, MA, USA), or 10 ng/mL of bFGF. On Day 4, half of the medium was replaced with a new differentiation medium. In order to investigate the effect of different factors on L-homocysteine (69453, Sigma-Aldrich, Saint Louis, MO, USA) toxicity, NSCs were divided into two groups: an FBS-treated group and a bFGF-treated group. Each of these two groups were further subdivided into three subgroups for L-Hcy treatment: control, a concentration of 300, and a concentration of1000 µM of L-homocysteine. As we wanted to follow changes over time, we observed this hierarchy on Day 3 and Day 7 of differentiation.

### 4.3. Immunocytochemistry

Cells were grown on coverslips and fixed in 4% PFA for 10 min and washed 3 times with PBS. Permeabilization was made in 0.2% Triton in PBS for 15 min and washed again 3 times with PBS. Cells were blocked in filtered 3% Goat serum in PBS, at room temperature for 2 h. Cell proliferation was analyzed with EdU Staining (Click-iT™ Plus EdU Cell Proliferation Kit for Imaging, Alexa Fluor™ 555 dye, C10638, Thermo Fisher Scientific, Waltham, MA, USA). Primary antibodies were added directly on coverslips and incubated overnight at +4 °C. Primary antibodies used were: anti-Nestin (MAB353, Merck, Darmstadt, Germany) 1:200, anti-TUBB3 (Biolegend, 802001, Biolegend, San Diego, CA, USA) 1:1000, anti-MBD1 (NV100-55537, Novus Biologicus, Littleton, CO, USA) 1:500, Anti-GFAP (ab4674, Abcam, Cambridge, MA, USA) 1:1000, anti-SOX2 1:200 (#23064, Cell Signaling, Danvers, MA, USA). Next day, cells were washed 3 times in PBS, after which secondary antibody was added and incubated at room temperature for 1 h. β−Actin was stained with Phalloidin–Atto 565 (94072, Sigma-Aldrich, Saint Louis, MO, USA). We used 1:1000 goat anti mouse 488, 1:1000 goat anti chicken 546 and 1:1000 goat anti rabbit 633. Cells were washed 3 times in PBS. Counterstaining was carried out with DAPI (1:20,000).

### 4.4. Pyknotic and Live Nuclei Demarcation in Imaris Software Version 9.9.1

We demarcated pyknotic and live nuclei under the assumption that condensed, smaller and more spherous nuclei represent pyknotic nuclei, based on DAPI as a nuclear stain. Firstly, we demarcated nuclei solely based on nuclei morphology using the Surface module in Imaris. By analyzing images, we discovered that nuclei negative for nuclear markers (SOX2 or MBD1) were pyknotic; therefore, we devised a method for demarcation of these two different nuclei in Imaris software version 9.9.1. We used the Surface module in Imaris on a nuclear marker channel to obtain nuclear marker-positive nuclei. Nuclear marker-positive nuclei Surface modules were then used to mask the DAPI channel, which resulted in two demarcated populations of nuclei: nuclear marker-positive nuclei named live nuclei, and nuclear marker-negative nuclei named pyknotic nuclei. This method was then further confirmed by measuring nuclei sphericity and nuclei volume.

### 4.5. Western Blot

Proteins in cell lysates, obtained by RIPA lysis buffer with added protease and phosphatase inhibitors, were sonicated for 10–15 seconds with 20–30% sonicator power (Qsonica CL188, Cole-Parmer, Vernon Hills, IL, USA). Protein concentrations were quantified by using bicinchonic acid method. The proteins were loaded into a 12% stain-free gel, and the blocking was performed using a 5% low fat milk for 1 h. The membranes were incubated overnight at +4 °C with the following antibodies: anti−TUBB3 (802001, Biolegend, San Diego, CA, USA)) 1:5000, anti−MBD1 (NV100-55537, Novus Biologicus, Littleton, CO, USA) 1:1000, anti−GFAP (ab4674, Abcam, Cambridge, UK), anti−SOX2 1:5000 (#23064, Cell Signaling, Danvers, MA, USA). The membranes were then washed three times in TBST and incubated with secondary antibodies (1:200,000) for 1 h. Subsequent detections were made using Femto SuperSignal chemiluminescent reagent (34095, Thermo Fisher Scientific, Waltham, MA, USA). The results obtained with the Western blot were normalized to the total protein amount. Images of full blots with loading controls can be found in the Appendix A.

### 4.6. DNA Methylation Analysis

DNA was isolated from adherent NSCs culture according to in-house protocol. Firstly, media above cells was aspirated and cells were washed with PBS, after which 200 µL of Lysis buffer was added (Tris 50 mM, pH = 8.0; EDTA 100 mM, pH = 8.0; NaCl 100 mM; SDS 1%; Proteinase K 0.2 mg/mL). Cells were scraped with a cell scraper and transferred to a tube. Cells in lysis buffer were agitated for 1 h at 56 °C at 1400 rpm in a thermoblock. DNA was precipitated using 200 µL isopropanol and gently inverted to form a precipitate. DNA was spun down for 10 min at 15,000× *g*. The DNA pellet was washed with 500 µL sterile 70% ethanol and inverted several times. The pellet was further spun down for 2 min at 15,000× *g,* and supernatant was carefully removed. The DNA pellet was dried at room temperature overnight to remove any residual ethanol. The next day, DNA was resuspended in mQwater. DNA concentration and quality were measured using a NanoDrop ND-1000 spectrophotometer (NanoDrop Technologies, Wilmington, DE, USA). Samples were then stored at −20 °C until further use.

All the procedures in this paragraph were carried out according to the manufacturer’s instructions. Some 500 ng of isolated genomic DNA underwent bisulfite treatment using an EpiTect Plus DNA Bisulfite Kit (#59124; Qiagen, Hilden, Germany). Then, 10 ng of bisulfite-treated DNA was used as a template for polymerase chain reaction (PCR) amplification of the promoter region of genes of interest using the PyroMark PCR Kit (#978703; Qiagen, Hilden, Germany). PCR protocol was as follows: initial denaturation at 95 °C for 15 min; 50 cycles of denaturation at 94 °C for 30 s, annealing at 58 °C for 30 s, and extension at 72 °C for 30 s; final extension was at 72 °C for 10 min. Biotinylated reverse primer was used. The biotinylated PCR product was purified using the Pyromark Q24 Vacuum Workstation (Qiagen). The *SOX2* and *TUBB3* promoters’ DNA methylation levels were measured using a Pyromark Q24 Advanced System with PyroMark Q24 CpG Advanced Reagents (#970922; Qiagen, Hilden, Germany) and a corresponding sequencing primer. DNA methylation levels were calculated as the ratio of C/T at a CpG site using the Pyromark Q24 Advanced Software 3.0.1 (#9022779; Qiagen, Hilden, Germany). Pyrosequencing conditions, used primers, and analyzed sequences are listed in Table 1.

### 4.7. Statistical Analysis

Statistical analysis was performed using R: A language and environment for statistical computing (R Foundation for Statistical Computing, Vienna, Austria. URL https://www.R-project.org/, version 4.1.3, accessed on 10 March 2022). All data were tested for normal distribution using the Shapiro–Wilk test, and parametric or non-parametric statistics were used as appropriate. Depending on the experiment design, we used a one-way ANOVA or a two-way ANOVA for parametric distribution, and Kruskal−Wallis for non-parametric distribution, followed by a Benjamini−Hochberg post hoc test for multiple comparisons.

## 5. Conclusions

In conclusion, here, we showed that supplementation of basal medium for differentiation of neural precursors with FBS and bFGF significantly influences not only basic cell parameters like their number, death and differentiation, but also influences how cells react to treatment with added molecules. In this case, we demonstrated that the choice of growth media supplementation affected how the cells responded to Hcy. In addition, DNA methylation analysis suggested that epigenetic profile was also influenced by growth medium supplementation.

## Figures and Tables

**Figure 1 ijms-24-14161-f001:**
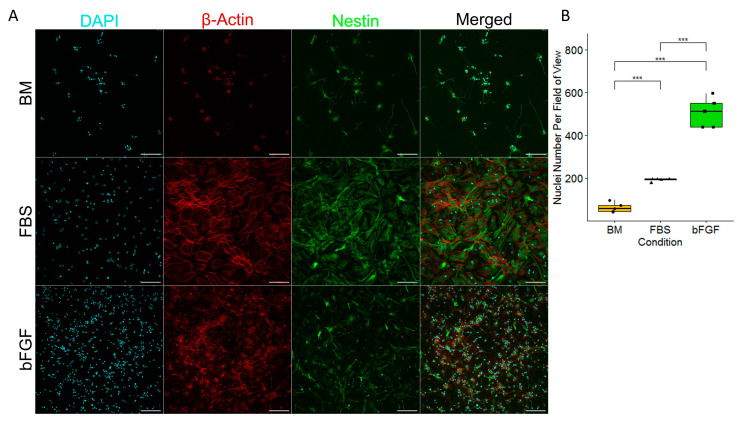
(**A**) Fluorescent microscopy of NSCs differentiated for 168 h (7 days) on 20× magnification. The scale bar is 100 µm. (**B**) Nuclei number per field of view. *n* = number of fields of view, graphically indicated by a dot for each point of observation. Significance level is indicated with *** for *p* < 0.001.

**Figure 2 ijms-24-14161-f002:**
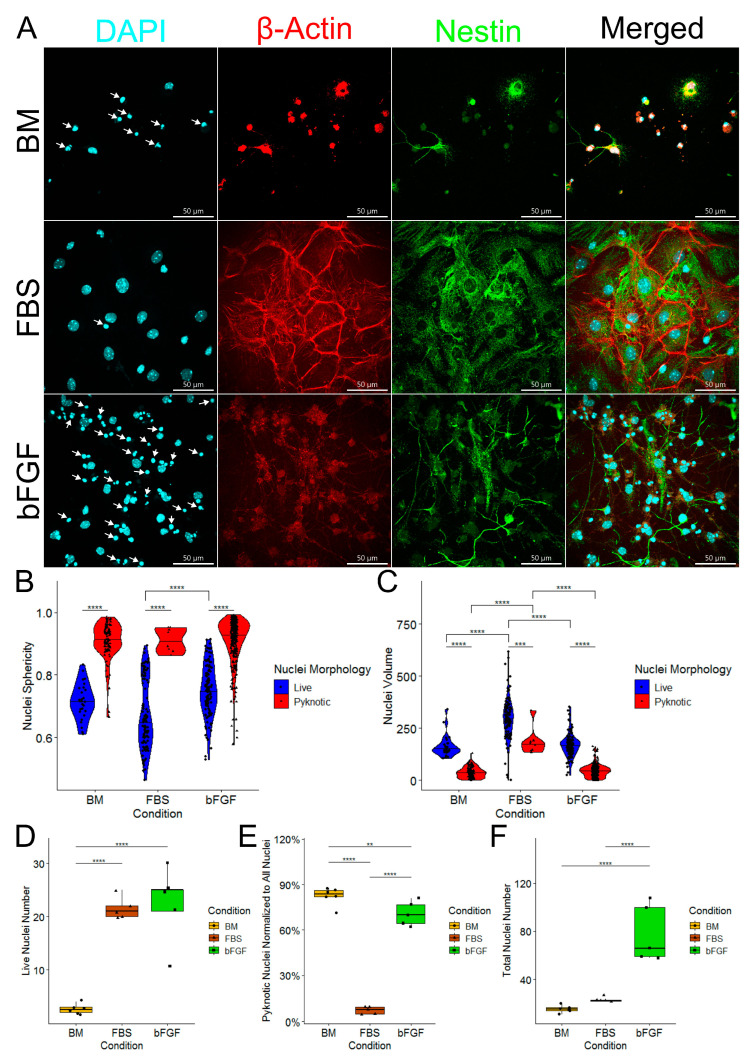
(**A**) Fluorescent microscopy of NSCs differentiated for 168 h (7 days) on 60× magnification. The scale bar is 50 µm. Pyknotic nuclei are marked with white arrows. (**B**) Nuclei sphericity. (**C**) Nuclei volume in µm^3^. (**B**,**C**) *n* = number of demarcated nuclei. (**D**) Live nuclei number per field of view. (**E**) Pyknotic nuclei normalized to total nuclei number (live + pyknotic nuclei). (**F**) Total nuclei number (live + pyknotic nuclei). (**D**–**F**) *n* = number of fields of view, graphically indicated by a dot for each point of observation. Significance levels are indicated with **, *** and **** for *p* < 0.01, *p* < 0.001, and *p* < 0.0001, respectively.

**Figure 3 ijms-24-14161-f003:**
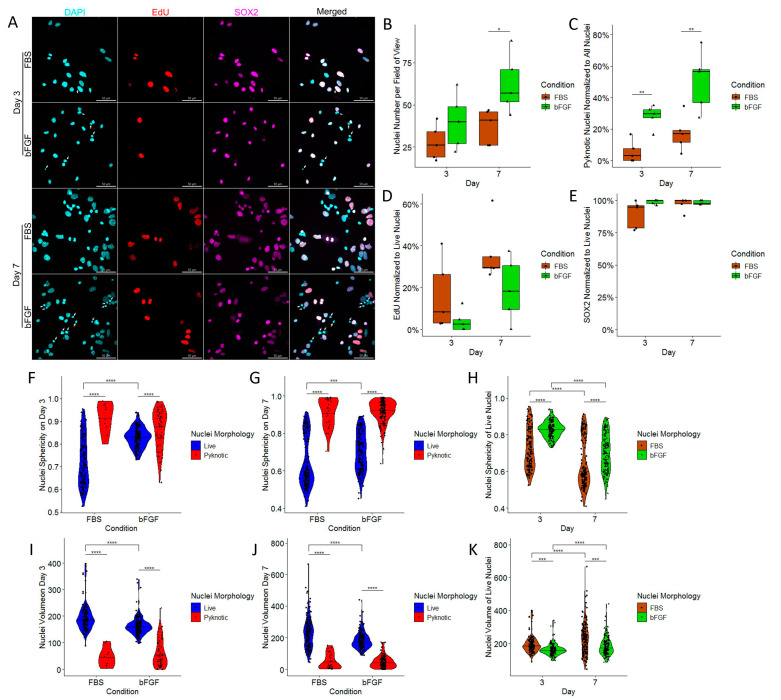
(**A**) Fluorescent confocal microscopy of NSCs differentiated for 72 h (3 days) in the top two rows, and for 168 h (7 days) in the bottom two rows. The scale bar is 50 µm. For each day, the top row represents images of NSCs treated with FBS, and the bottom row represents NSCs treated with bFGF for corresponding day. (**B**) Total number of nuclei per field of view. (**C**) Pyknotic nuclei normalized to all nuclei. (**D**) EdU-positive nuclei normalized to live nuclei. (**E**) SOX2-positive nuclei normalized to live nuclei. For (**B**–**E**), *n* = field of view, graphically indicated by a dot for each point of observation. Significance levels are indicated with * and ** for *p* < 0.05 and *p* < 0.01, respectively. (**F**) Nuclei sphericity on Day 3. (**G**) Nuclei sphericity on Day 7. (**H**) Live nuclei sphericity on Day 3 and Day 7. (**I**) Nuclei volume on Day 3. (**J**) Nuclei volume on Day 7. (**K**) Live nuclei volume on Day 3 and Day 7. For figures (**I**–**K**) volume in µm^3^. For (**F**–**K**) *n* = number of demarcated nuclei, graphically indicated by a dot for each point of observation. Significance levels are indicated with *** and **** for *p* < 0.001, and *p* < 0.0001, respectively. Pyknotic nuclei are highlighted with white arrows.

**Figure 4 ijms-24-14161-f004:**
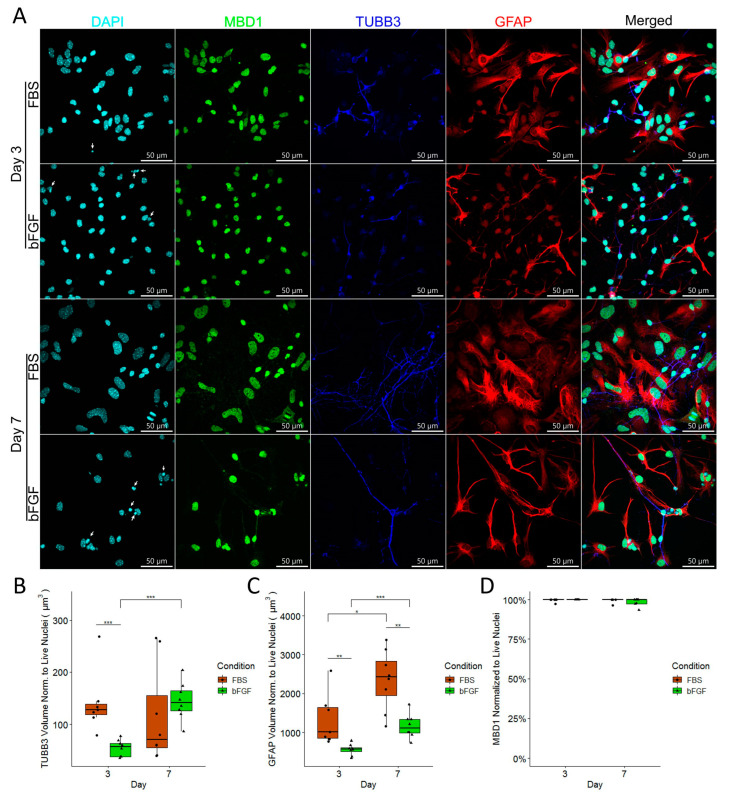
(**A**) Fluorescent confocal microscopy of NSCs differentiated for 72 h (3 days) in the top two rows and for 168 h (7 days) in the bottom two rows. The top row represents Figures of NSCs treated with FBS, and the bottom row represents NSCs treated with bFGF for the corresponding day. Pyknotic nuclei are highlighted with white arrows. (**B**) TUBB3 volume normalized to the number of live nuclei. (**C**) GFAP volume normalized to the number of live nuclei. (**D**) MBD1-positive nuclei normalized to the number of live nuclei. For (**B**–**D**), *n* = field of view, graphically indicated by a dot for each point of observation. Significance levels are indicated with *, ** and *** for *p* < 0.05, *p* < 0.01, and *p* < 0.001, respectively.

**Figure 5 ijms-24-14161-f005:**
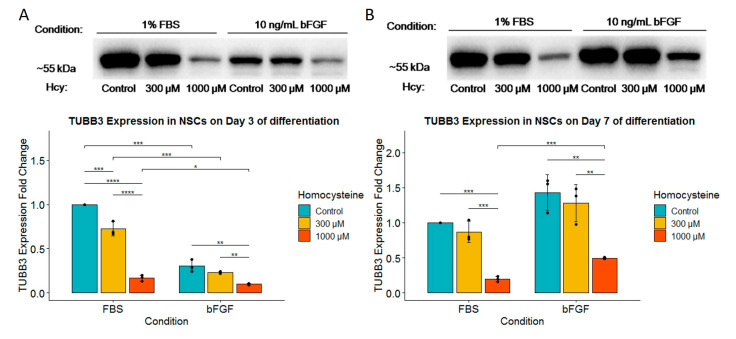
Western blot of TUBB3 (**A**) Western blot of TUBB3 expression for Day 3 of differentiation with corresponding fold change for TUBB3 expression underneath; *n* = 3 independent experiments, graphically indicated by a dot for each point of observation. (**B**) Western blot of TUBB3 expression for Day 7 of differentiation with corresponding fold change for TUBB3 expression underneath, *n* = 3 independent experiments, graphically indicated by a dot for each experiment. Significance levels are indicated with *, **, *** and **** for *p* < 0.05, *p* < 0.01, *p* < 0.001, and *p* < 0.0001, respectively.

**Figure 6 ijms-24-14161-f006:**
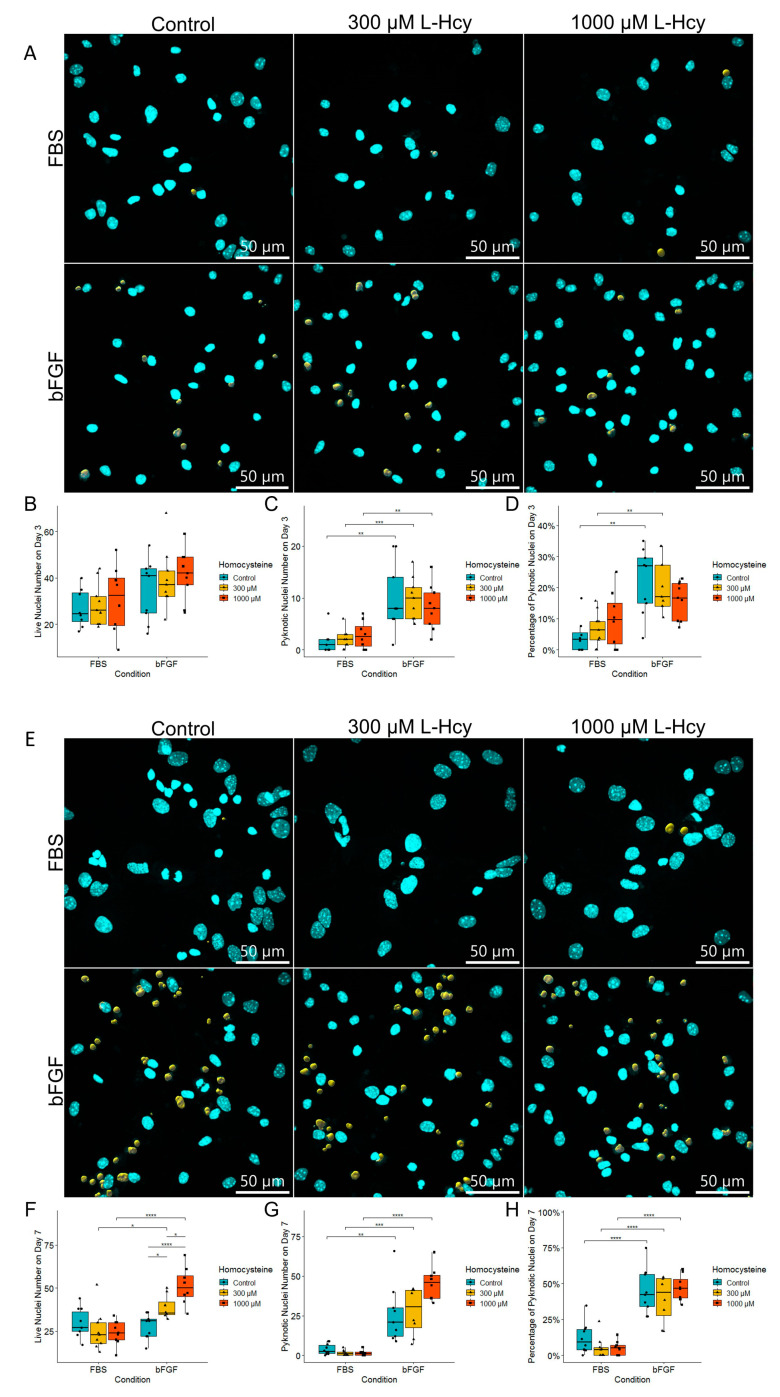
Fluorescent confocal microscopy of NSCs differentiated for 72 h (3 days) (**A**) and for 168 h (7 days) (**E**). The top row represent NSCs treated with FBS, and the bottom row represents NSCs treated with bFGF. Pyknotic nuclei are marked in yellow. (**B**) Live nuclei number on Day 3. (**C**) Pyknotic nuclei number on Day 3. (**D**) Percentage of pyknotic nuclei on Day 3. (**F**) Live nuclei number on Day 7. (**G**) Pyknotic nuclei number on Day 7. (**H**) Percentage of pyknotic nuclei on Day 7. (**B**–**D**,**F**–**H**): *n* = field of view, graphically indicated by a dot for each point of observation. Significance levels are indicated with *, **, *** and **** for *p* < 0.05, *p* < 0.01, *p* < 0.001, and *p* < 0.0001, respectively.

**Figure 7 ijms-24-14161-f007:**
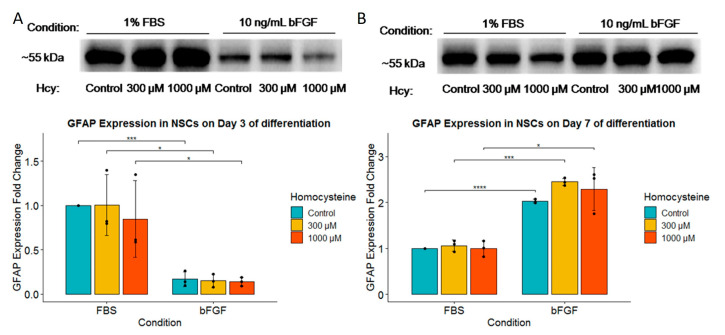
Western blot of GFAP. (**A**) Western blot of GFAP expression for Day 3 of differentiation with corresponding fold change for GFAP expression underneath; *n* = 3 independent experiments, graphically indicated by a dot for each point of observation. (**B**) Represents Western blot of GFAP expression for Day 7 of differentiation with corresponding fold change for GFAP expression underneath; *n* = 3 independent experiments, graphically indicated by a dot for each experiment. Significance levels are indicated with *, *** and **** for *p* < 0.05, *p* < 0.001, and *p* < 0.0001, respectively.

**Figure 8 ijms-24-14161-f008:**
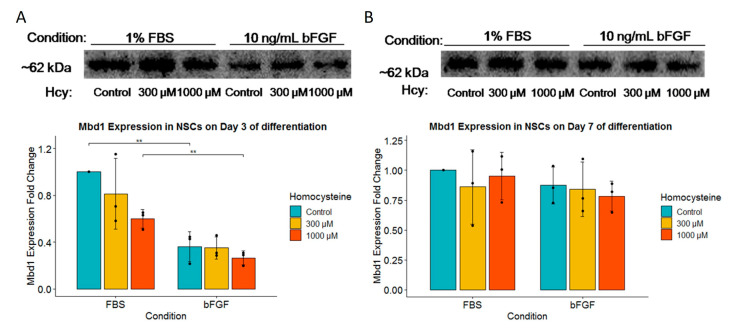
Western blot of MBD1. (**A**) Western blot of MBD1 expression for Day 3 of differentiation with corresponding fold change for MBD1 expression underneath; *n* = 3 independent experiments graphically indicated by a dot for each point of observation. (**B**) Western blot of MBD1 expression for Day 7 of differentiation with corresponding fold change for MBD1 expression underneath; *n* = 3 independent experiments, graphically indicated by a dot for each experiment. Significance level is indicated with ** for *p* < 0.01.

**Figure 9 ijms-24-14161-f009:**
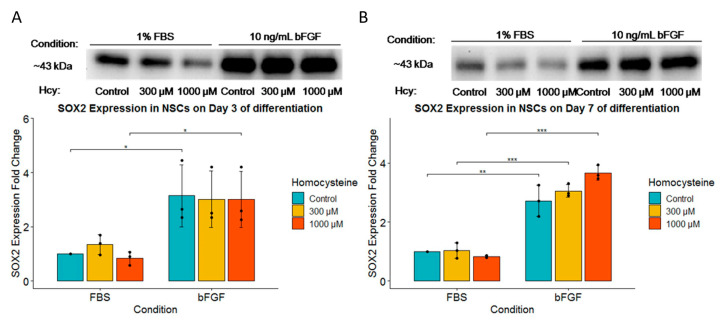
Western blot of SOX2. (**A**) Western blot of SOX2 expression for Day 3 of differentiation with corresponding fold change for SOX2 expression underneath; *n* = 3 independent experiments, graphically indicated by a dot for each point of observation. (**B**) Western blot of SOX2 expression for Day 7 of differentiation with corresponding fold change for SOX2 expression underneath; *n* = 3 independent experiments, graphically indicated by a dot for each experiment. Significance levels are indicated with *, **, and *** for *p* < 0.05, *p* < 0.01, and *p* < 0.001, respectively.

**Figure 10 ijms-24-14161-f010:**
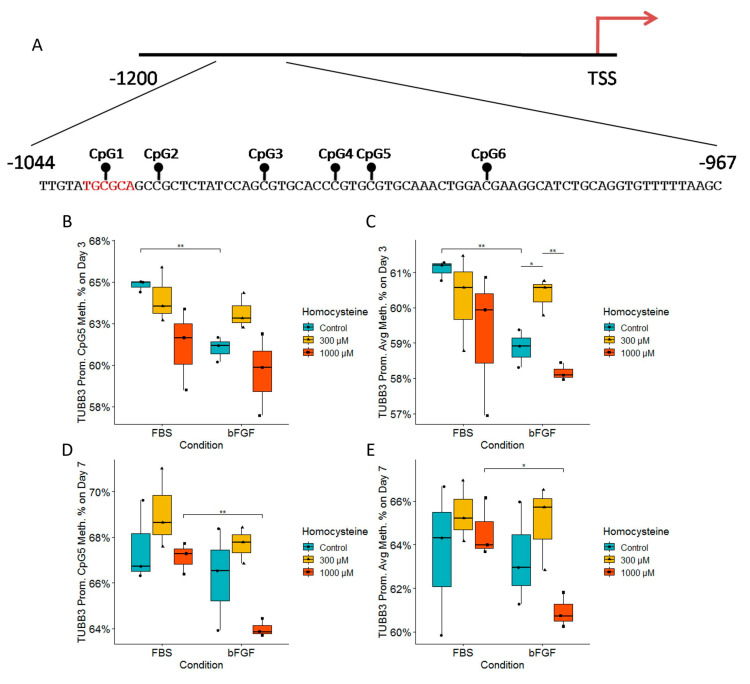
DNA methylation of the *Tubb3* and *Sox2* promoters during neural stem cells’ differentiation. (**A**) Structure of the *Tubb3* promoter. The black lines and circles on the diagram each represent CpG dinucleotide with assumed DNA methylation in a sequence region upstream of the transcription start site (TSS); the preferential binding site for MBD1 is marked in red. (**B**) CpG5 dinucleotide DNA methylation for the *Tubb3* promoter on Day 3 of differentiation. (**C**) Average DNA methylation of all six CpG dinucleotides of the *Tubb3* promoter on Day 3 of differentiation. (**D**) CpG5 dinucleotide DNA methylation of the *Tubb3* promoter on Day 7 of differentiation. (**E**) Average DNA methylation of all six CpG dinucleotides in the *Tubb3* promoter on Day 7 of differentiation. (**F**) Structure of the *Sox2* promoter. The black lines and circles on the diagram each represent CpG dinucleotide with assumed DNA methylation in a sequence region upstream of the transcription start site (TSS). (**G**) CpG2 dinucleotide DNA methylation of the *Sox2* promoter on Day 3 of differentiation. (**H**) CpG9 dinucleotide DNA methylation of the *Sox2* promoter on Day 3 of differentiation. (**B**–**E**,**G**–**H**); *n* = 3 independent experiments, graphically indicated by a dot for each experiment. Significance levels are indicated with *, and ** for *p* < 0.05, and *p* < 0.01, respectively.

**Table 1 ijms-24-14161-t001:** This table represents primers used for bisulfite-treated DNA PCR and Pyroseqencing.

Gen	Primer	Sequence	bp	Annealing Temp.	Number of Cycles
TUBB3	F	5′-GGGGAGGGGTATTTTTTGAGAATAA-3′	199	56 °C	50
	R	5′-BIOTIN-ACACTTCAATAATCCACAAACATT-3′			
	Seq	5′-GGATGTTTATTTTTAGAGAAAG-3′			
	Sequence to analyze	5′-TTGTATGYGTAGTYGTTTTATTTAGYGTGTATTYGTGYGTGTAAATTGGAYGAAGGTATTTGTAGGTGTTTTTAAGT-3′			
SOX2	F	5′-GTTTGGGTTTGTTTGGTG-3′	249	56 °C	50
	R	5′-BIOTIN-AACTTCCTAACATCCCAC-3′			
	Seq	5′-TGTATTTGTATTTTTGG-3′			
	Sequence to analyze	5′-ATTTYGYGYGTTTTTTAGGTTTYGGYGTTTTTYGGTYGGGTTTTYGTGATTATTATYGTGTTTGTTAGTAGGG-3′			

## Data Availability

All data are available upon reasonable request.

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
