# Peer review of "Effect of Fetal Bovine Serum or Basic Fibroblast Growth Factor on Cell Survival and the Proliferation of Neural Stem Cells: The Influence of Homocysteine Treatment"

_ijms, 2023, doi:10.3390/ijms241814161_

Round 1
Reviewer 1 Report
Dear Editor-in-Chief (IJMS, MDPI)
Supplementation of neural stem cells with fetal bovine serum or basic fibroblast growth factor influences cells` survival, proliferation and response to treatment by homocysteine by Dražen Juraj Petrović et al
I propose this better title ¨ Effect of fetal bovine serum or basic fibroblast growth factor on cell survival, proliferation of neural stem cells: influence of homocysteine treatment¨.
This study clearly show how different treatment (trophic factors, FBS, médium) can influence the survival of immature precursors of NSC, depending of growth conditions with different addd factors to the médium. TIn this study, nuclei sphericity, picknotic nucleus or volumen were evalute among different treatments in NSC. This is a really important study but with methological errors.For example, control load are missing in western blots for GFAP , Sox-2, which means these attributable differences to different treatments could also attributable to different load simples. Please, shall you show western blot for load controls (actine, beta-actinine , GDPH, etc) ? in your study in order to solve this problem.
The begining of discusion should explain better all differences between FBS, media and FGF tratements in terms of volumen nuclei, nuclei sphericity and picknocitic cells by all these treatments and how NSC can be differentiate to neurons/astroglia depending of each treatment. Please, also indicate the number of seeded cells by experiment (figures 1 to 4). Althouth this is really very interesting study, I miss load controls in western blot experiments, which is a important methodological problem in this study. If molecular weigh of actine is similar (43 Kda) to some marker here, you can use stripping on membranes or another control load marker such as actinine or Pi-kinase subunits, GAPH, etc). I also miss nuclear control load for western of TUBB3.
-The begining of discusión should be improved by explaining all differences in terms of picknotic nucleus, volumen nucleus, as well as differential capacities of NCS differentiation to neurons/astrocytes depending of FGF and FBS treatments. In addition, compare your findings with published evidences in the field of NSC differentiation to neurons/glia. Although homocystein take relationship with neurodegeneration, these data can not be extended to neurological diseases without evidences in the manuscrip by these authors.
--Extensive editing of English language is required
-These references are not in IJMS (MDPI) format. Please, format these references to the format of this Journal.
-Please, also include a conclusion in your study.
My Decision is Major revision. I can not accept this study with minnor revision by these arguments but is also excesive the consideration of Major revision (controls missing and allow its resubmission). If authors show control load for western blot, my Decision turn to minnor revision but without these control load in western blot, my profesional Ethics does not allow me although Major revision is the real state of this evaluation. I would like to notice that Major revision only has dissaper in MDPI evaluations. I press on line Major revision (resubmission) because the system does not allow Major revision only.
The references are in another format, which is different to IJMS (MDPI).
Thanks
Minnor comments
Dear authors
-They use the term cultivation of cells although the term growth of cells, culture cells (in vitro) is more appropiate and more accepted by researchers.
-Line 27. ¨ Here, we have clearly shown that selection of culturing media significantly influence effects measured on in vitro models of neurological diseases, which in turn can lead to different conclusions in research setups using in vitro models¨. In my opinion, these findings can not be extrapolated to neurological diseases because their study did not evaluate markers of neurodegeneration in patients¨
Remove this part within the introduction because these parameters have not been tested in your study. Thus, remove these lines from ¨Prevailing view of mechanistic cause for Alzheimer's disease (AD) today is the amyloid hypothesis. The amyloid hypothesis of AD is centered around deposition of amyloid beta which progressively leads to accumulation of Tau, synaptic dysfunction, inflammation, neuronal loss, and, ultimately, dementia [3],
which is as well shown in an in vitro model used by our group [4].¨ This text is not focus on the aim of this study without evaluating markers of neurodegeneration such as amyloid beta or Tau levels¨
Line 93. Repeated twice this sentence. ¨ Number of Nuclei per field view. n = number
- Why does you recognize these more spherous nucleus as Pkynotic nuclei (Figure 2.A - white arrow.). I understand the meaning of brigth and small nucleus as pycnotic. Please, explain this criterio of selection for pycnotic analisis.
-line 102. How do you analyze/quantify difference in sphericity and volumen between live and pyknotic nuclei¿ Did you use methamorph sowftware or J image? Did you measure the distance from the nucleus to the periphery of each cell?
After revising, figure 2A, the, the FBS médium is the best for culture NSC since there was many less pincnotic nucleus although there is less live nucleus as compare to FGF treatment? In additon, the fluorescence of figure 2 A suggest that nestine colocalizes with neurons by FGF recombinant treatmen while FBS did not lead to neuron phenotype differention of these culture NSC in vitro. Please, shall you expain this discrepance in terms of NSC differentiation towards neurons by these differential treatments?
- line 174. Please, explain the reason by which nuclei sphericity was significantly reduced on day 7 compared to day 3 in both FBS and bFGF.
-Nuclear control load for western Blot of TUBB3 are missing in this study as well as control load for another evaluated proteínas such as Sox-2, and GFAP, which were not showed by authors. It is correct to compare the intensity bands as compare to controls but with normalization with control loads. However, control load for proteins are missing in all the study (actine, actinine, etc). Sox-2 western blot has a similar weigh that actine (43 Kda) but in this case it is also valid a stripping of the membrane for testing actine again even you can lost protein. The other option is to search another load control different of 43 Kda (figure 9) for normalization in western blot.
-Line 438-439. -In vitro models are a very commonly used tool in neuroscience. Thus there are many pub- 437 lished papers reporting successful application of in vitro model of various neural cells in 438 Amyotrophic lateral sclerosis ALS [16], Alzheimer`s disease [17,18], Multiple Sclerosis [19] 439 and Parkinson Disease [20¨. Please, remove this sentence because you are not evaluating these models in vitro since these NSC are not subjected to abeta amyloid and these -pathologies are not studied here.
-Please, explain your data and differences between FGF and FBS treatments in NCS and differential differentiation to neurons/glia cell with published findings in the field.
-line 502. Explain, the reason by whihc just 1% of FBS also leads to development of reactiv eastrocyte morphology compared to bFGF 503 treated astrocytes in the discusion.
-line 530. These are not supplements . FGF or FBS are treatment but not really supplements. Please, correct it.
-Please, also include the conclusión in your study
The references are in another format, which is different to IJMS (MDPI).
-Extensive editing of English language is required
My Decision is Major revision. My Decision is Major revision. The absence of control loads for western blot forces me to take that decision although the word is really interesting. In this way, major revision only as decision has dissapaer within MDPI system and now include resubmission.
Thanks¡
-Extensive editing of English language is required
The references are in another format, which is different to IJMS (MDPI).
Reviewer 2 Report
This is an interesting work showing the effects of different culture media on some biological activity of neuronal stem cells. The results are interesting but overall the manuscript need deep revision to make the manuscript more effective and easy to follow and understand. Some suggestions are as follow:
The abstract is too general; it should clearly state the observed effects. In other words, it is not enough to report "were significantly affected" (line 23), but it should state whether there was a decrease or increase in biological activity.
The introduction should be more incisive and well organized into several separate sections as follows: 1) state of the art; 2) report clearly what the authors has done in previous work; 3) report briefly what is the objective of the present work; 4) reduce/eliminate the description of the results from the introduction. (Lines 66-76: this part should not be in the introduction section).
Figure 1B: the legend is not necessary
Line 151: please explain the meaning of “n = number of demarcated nuclei”
Line 188: please explain the meaning of “n = Field of view”
Lines 197-210: why Figure 4C is described before Figure 4A? This part is not clear
Lines 249-257: why Figures 6A, 6B, 6C and 6E have not been mentioned
Line 600: explain in more detail the NCI isolation procedure or add a reference. Similarly, in the other methods section, include a reference if the same procedures have been used before.
The all results section need a deep revision.
Round 2
Reviewer 1 Report
Dear authors
All requirements were solved by authors. Additionally, the initial concern about control load in western blot has been satisfactory solved by these authors. Their response to my question about the lack of control load for western blot study is satisfactory for me. The textually indicate ¨Response 2. Since we have compared normalizations with total lane protein to approach over β-Actin and we found that there was no difference, we normalized our blots to the total lane protein. Additionally, many published papers demonstrate that stain free technology is superior to housekeeping proteins and even recommended for western blot quantification (see: https://doi.org/10.1155/2014/361590 and¨https://doi.org/10.1016/j.ab.2012.10.010
Thanks¡
Please, correct typos errors.
Author Response
Comment 1. Please, correct typos errors.
Respond 1. We thank the reviewer 1 for pointing out typos errors. We have corrected typos errors.
Reviewer 2 Report
None
Author Response

(The authors gave the same response as above.)
